# Rural General Practitioners’ Perceptions of the Barriers and Facilitators of Chronic Disease and Cardiometabolic Risk Factor Care Through Lifestyle Management—A Western Australian Qualitative Study

**DOI:** 10.3390/healthcare14010113

**Published:** 2026-01-02

**Authors:** Aniruddha Sheth, Sandra C. Thompson, Nahal Mavaddat

**Affiliations:** 1Discipline of General Practice, UWA Medical School, University of Western Australia Discipline of General Practice, 35 Stirling Highway, Crawley, WA 6008, Australia; 2Western Australian Centre for Rural Health, School of Allied Health, University of Western Australia, P.O. Box 109, Geraldton, WA 6531, Australia

**Keywords:** barriers, cardiometabolic risk factors, chronic disease, facilitators, general practitioners, lifestyle management, metabolic syndrome, qualitative research, rural health, Western Australia

## Abstract

Background: Chronic diseases such as type 2 diabetes mellitus and cardiovascular disease and their cardiometabolic risk factors require management, which includes lifestyle interventions. Rural and remote residents are disproportionately affected by these conditions compared to their urban counterparts. Studies have examined barriers to chronic disease and cardiometabolic risk factor management in urban environments, but rural perspectives remain underexplored, especially in Western Australia (WA) with its vast geography. This study examined rural general practitioners’ (GPs) views on barriers and facilitators to chronic disease and cardiometabolic care in rural WA through lifestyle management. Methods: This qualitative study used semi-structured interviews with 15 rural WA GPs recruited via rural networks using convenience and snowball sampling. Braun and Clarke’s reflexive thematic analysis was used to identify patterns and themes within the qualitative data that addressed the study questions. Results: According to rural general practitioners, major barriers to chronic disease and cardiometabolic risk care included geographic isolation, socioeconomic disadvantage and an obesogenic food environment in rural areas, as well as severe time and financial constraints for GPs and workforce shortages with a high turnover and lack of accessible allied health professionals. Facilitators included co-located multidisciplinary teams, case management/health coaching, better remuneration for complex consultations involving preventive care and upstream policy measures, such as improving healthy food affordability and availability. Conclusion: Rural patients face systemic, geographic and socioeconomic barriers that are substantially greater than those in urban settings; these barriers impact GPs caring for their patients with chronic disease and cardiometabolic risk factors. Targeted solutions to these barriers such as attention to workforce issues, investment in lifestyle coaching approaches and having dedicated case managers, could reduce rural–urban inequities in chronic disease outcomes.

## 1. Introduction

Chronic diseases such as diabetes (T2DM) and cardiovascular disease (CVD) are leading contributors to morbidity and mortality worldwide and place a substantial burden on healthcare [1]. In Australia, these conditions account for a significant proportion of the disease burden, particularly in rural and remote areas, where there is often limited access to healthcare services [2] and a high prevalence of conditions that predispose people to chronic diseases [3]. Obesity is common in rural Australia [4], as are other cardiometabolic risk factors such as hypertension, dyslipidaemia and impaired glucose metabolism [5,6,7,8,9]: all key components of the so-called metabolic syndrome [10]. In Australia, estimates suggest 20% to 30% of adults are affected by a combination of these cardiometabolic risk factors [10]. In rural areas, these rates can be even higher [4,10]. The burden of risk is also often heightened in rural settings by delayed diagnosis and treatment of these predisposing factors, leading to higher rates of complications of T2DM and CVD than in urban settings [11,12]. Current management of these risk factors includes medications and surgery and also centres on addressing lifestyle factors and promoting dietary and exercise interventions [5,13,14]. Western Australia is 2,527,013 square kilometres, which is close to a third of Australia’s land mass, but it contains only 11% of the Australian population; the small numbers of people living over this vast area challenge the equitable provision of healthcare [15].

General practitioners (GPs) in rural Australia are central to the management of chronic conditions including T2DM and CVD. GPs in rural areas often serve as the primary, and sometimes only, healthcare providers who manage these conditions, especially in isolated rural communities [16]. However, rural GPs face significant barriers to delivering chronic disease care, and to accomplishing the lifestyle changes that are key to improving outcomes for their patients. Such barriers to providing effective chronic disease care and lifestyle management, in addition to geographical isolation, include healthcare workforce shortages in rural areas and, especially, poorer access to allied health practitioners [16,17,18]. However, facilitators of good chronic disease care and lifestyle management to control cardiometabolic risk remain underexplored in the context of rural Australia. Indeed, while barriers and facilitators to chronic disease and cardiometabolic risk factor management have been studied in some urban populations [19]—for example, the struggle to meet targets with limited time and resources, they have not been extensively examined within the rural environment. Rurality adds an additional dimension to the challenges for practitioners in supporting patients with chronic disease and lifestyle management.

Chronic disease diagnosis and therefore management is frequently delayed in rural compared to urban Australia, with rural populations having a higher disease and risk factor burden. Western Australia includes regions with sparse populations, including isolated Aboriginal populations with greater risks of chronic disease and with specialised health needs [20]. Rural patients often experience poorer service availability and higher costs of access to healthcare, which compound care for chronic conditions and contribute to poorer health outcomes [21]. Much of the existing literature on rural healthcare draws from national data or data from states where rural areas often have denser populations and are closer to metropolitan centres [2,22]. In contrast, Western Australia presents distinct challenges due to its vast geography and dispersed and sometimes sparse populations, with remote communities located thousands of kilometres from Perth, its urban centre. Service distribution is often more centralised, with referral pathways being in a hub and spoke models frequently being centralised to Perth [23], with reliance on outreach, fly-in/fly-out locum providers, nurse-led services, telehealth services and the Royal Flying Doctor Service (RFDS) for very remote areas [22,24]. These unique features of WA exemplify the intensified challenges of rural healthcare delivery, making it critical to explore the context of rural GPs’ perceptions of cardiometabolic care via lifestyle management.

We have recently published on the perspectives of GPs in their approach to the cardiometabolic risk factors of chronic conditions such as T2DM and CVD in the context of metabolic syndrome [25]. In this paper, we outline in greater detail our findings on the barriers and facilitators to chronic disease care and cardiometabolic risk factor management, particularly in relation to lifestyle changes in the context of rurality. We have utilised qualitative reflexive thematic analysis of semi-structured interviews with rural GPs. By identifying the specific barriers that rural GPs encountered and the facilitators to better management of cardiometabolic risk, this research provides actionable insights for enhancing chronic disease management in rural settings. The findings can inform targeted interventions and policies that better support GPs, improve healthcare delivery and reduce health disparities between rural and urban populations in Australia.

## 2. Materials and Methods

This study involved semi-structured interviews with GPs in rural Western Australia. Full methods are presented elsewhere [25]. The present analysis examined GPs’ perspectives of the barriers and facilitators to chronic disease care in rural Western Australia.

The research team comprised AS, a GP registrar at the time of the study with an interest in lifestyle interventions, and NM and ST, who are practitioners experienced in general practice and rural and public health, respectively. The interview guide was developed by the research team (AS, ST and NM) and informed by the relevant literature and was piloted initially with a rural GP (Appendix A).

### 2.1. Recruitment

Any interested GP or GP registrar working in rural WA was eligible for the study. There were no other exclusion criteria.

Recruitment utilised both convenience and snowball sampling. GP practices in the region were identified and invitations to participate were circulated through rural GP networks. Individual practices were approached directly or referred by others. Interested general practitioners received participant information sheets, consent forms and a link to a brief pre-interview survey.

### 2.2. Data Collection

Demographic information was collected at the time of the pre-interview survey.

AS conducted all interviews (between 45 and 60 min), either in person or via Zoom/Microsoft Teams, and all were audio recorded. Interviews proceeded as per the interview guide in Appendix A, focusing on how rural general practitioners’ perceptions regarding cardiometabolic syndrome and barriers and facilitators caused them to care for patients with cardiometabolic syndrome and cardiometabolic risk factors in the rural WA context.

### 2.3. Analysis

Transcripts were generated using Otter.ai software (Mountain View, California, USA, https://otter.ai/ (accessed on 20 December 2025), deidentified and manually verified for accuracy via verbatim editing and returned to participants for optional review and amendments.

NVivo software (Lumivero, Melbourne, Victoria, Australia release 1.2.1 (1534)) was used for data management. Analysis proceeded concurrently throughout data collection iteratively. Interviews continued until thematic saturation was achieved. Reflexive thematic analysis, as per Braun and Clarke’s methodology, was used to identify patterns of meaning in the collected interviews [26].

The primary coding was performed by AS through repeated transcript immersion, initial code generation and progressive refinement using Braun and Clarke’s methodology. A subset of interviews was separately coded by researcher NM with high concordance and the remaining discrepancies were resolved through a discussion with ST to finalise themes.

Thematic mapping involved transferring codes to a digital canvas tool (Concepts App TopHatch, Inc. San Carlos, California, USA version 2025.12.4) for visual organisation, grouping and exploration of interconnections [26]. An inductive, iterative approach identified overarching themes and their interrelationships, supported by an ongoing team dialogue to construct a coherent narrative that was grounded in the data.

To enhance rigour and to maintain trustworthiness, researchers undertaking the coding reflected on core themes via reflexive dialogue. Bracketing was used to ensure that the researchers were aware of their own interests and remained unbiased by their personal perspectives.

Ethics approval was obtained from a university human research ethics committee.

## 3. Results

Fifteen GPs were interviewed: nine male and six female. Twelve GPs were consulting part-time, three were practice owners and there was one GP registrar (GP in training). The age of the GPs ranged from 30 to 60 years, with a range of 0–30 years practicing post-fellowship in general practice. There were no refusals for interviews among the GPs that were personally approached.

Identified barriers and facilitators to providing chronic disease care through lifestyle management to patients with cardiometabolic risk factors in the rural Western Australian environment are described below and summarised in Figure 1.

### 3.1. Barriers to Care

#### 3.1.1. Systemic Challenges in Rural Healthcare Delivery for Chronic Disease and Metabolic Risk: Resource, Geographic and Socioeconomic Constraints Impacting on Care

Rural GPs reported encountering multiple systemic barriers—resource limitations, geographic isolation and socioeconomic disadvantage—that aggregated to impede their ability to deliver effective care for chronic disease and chronic disease risk factors which required lifestyle changes for a range of cardiometabolic risk factors. These challenges reflected the complex and often fragmented healthcare landscape for chronic disease risk factor management in rural Western Australia. Specifically, the GPs reported the following.

##### Time and Financial Constraints for Lifestyle Interventions and Advice

Scarcity of time to manage and provide lifestyle care to patients with chronic disease and cardiometabolic risk factors emerged as a pervasive barrier. GPs reported struggling with the demands of management for cardiometabolic risk in their patients within the confines of a 10 min consultation window. This required the GP to manage one risk factor over another and often was at the expense of comprehensive chronic disease management. These challenges to time allocation occurred partly due to an inadequate number of GPs, insufficient resources in rural communities and the Medicare billing system. One GP articulated this dilemma:

“In 10 min, I’m trying to think of, should I be helping this person with diabetes management today? Or should I be looking at blood pressure… it’s trying to prioritize what’s most important at [the] time.”(GP02)

Another GP noted the problem of time constraints in the context of providing comprehensive chronic disease care management:

“So, you have to choose what you’re telling them, what’s important that you want [to get] across to them… and then save the rest.”(GP03)

GP 14 described “the lack of even see[ing] a GP to start with…people who don’t really do checkups. They come when they’re sick. And so…we’re working on their illness rather than focusing on prevention of anything.” This was compounded by the lack of GPs, where “less medical students are wanting to be GPs… we are time strapped and having to, you know, even now turn away patients from practices in places like Geraldton.” (GP04).

Rural GPs also noted that challenges surrounding financial remuneration compounded those related to time pressures. The Australian fee-for-service model, funded largely through Medicare, prioritises shorter consultations; this creates tensions between thorough comprehensive care and ensuring the practice’s financial viability. This financial disincentive discouraged longer consultations that might allow for detailed lifestyle counselling or care planning, which are critical components of chronic disease management and management of metabolic and chronic disease risk. One GP explained the need to understand nuances of the Medicare billing system for funding, and the challenge to do the best for the patient, which would require a 20 min appointment, while “clinic viability is always in the back of your head.” (GP06).

##### Insufficient Allied Healthcare Resources

GPs reported difficulties with accessing the required care for their patients with chronic disease risk factors for lifestyle change, diet and exercise due to insufficient allied healthcare professional resources. Allied health support was regarded as being crucial for addressing chronic disease risk factors, but support was limited by the lack of availability of allied health practitioners and travel distance to appointments. This scarcity of support created challenges to managing chronic conditions holistically through using multidisciplinary teams which included dieticians and exercise physiologists. Staffing shortages, driven by high turnover and recruitment difficulties, impacted the comprehensive care of patients with cardiometabolic risk and who required chronic disease care. GPs reported that higher living costs and isolation of rural areas in WA exacerbated the ongoing struggle to attract and retain healthcare professionals.

One GP remarked as follows:

“I think getting people to live here, this is a hard place to live… it must probably be about 30 to 50% more expensive to live in here compared to Perth.”(GP14)

Another GP highlighted the broader implications, suggesting that due to the workforce shortage, allied health practitioners were overloaded “dealing with the secondary stuff, dealing with the fallout” (GP07), rather than engaging patients in preventive practices.

“It’s also just availability then of dieticians… they’re not available. Getting into appointments, it’s hard with transport, like so many things.”(GP10)

Communication between GPs and allied healthcare professionals was also identified as a barrier:

“It does feel sometimes when you send patients off to see allied health professionals that they just disappear into a void, like you don’t always get a letter back from them…”(GP05)

##### Geographic Isolation and Access to Healthcare in Rural Areas

These issues were all compounded and interacted dynamically with the geographical isolation of patients from practitioners, so that the environment for delivering optimal care for chronic condition risk factors became a formidable task for many. GPs reported that geographical isolation meant that not only were patients far from their surgery, which constrained patient attendance, but this also limited their access to allied health professional care.

The vast distances that are characteristic of rural Western Australia posed significant logistical challenges for patients to see practitioners. GPs reported that patients often faced difficulties accessing healthcare services due to transportation barriers, particularly in remote areas where public transport is scarce. A couple of GPs highlighted this issue:

“If they can drive, if they can’t drive… someone can drive them here, can’t bring them here. That is an ongoing issue.”(GP03)

“It’s also just availability of dieticians, physiotherapists or exercise physiologists…It’s not available. Get[ting] into appointments, it’s hard transport…remote communities where I work, they’re a couple of 100 kilometres from here and kind of narrow…like those people visit is just lots of things.”(GP10)

#### 3.1.2. Broader Challenges in Rural Areas Beyond Healthcare for Chronic Disease and Cardiometabolic Risk

Environmental pressures heightened the tension between individual responsibility and systemic influences on achieving good health outcomes, complicating patient-centred care delivery in the rural environment for patients with metabolic risk.

##### Challenges for Healthy Living in the Rural Environment

The food and economic system in the rural environment, for example, emerged as a powerful external barrier to chronic disease and cardiometabolic risk factor management, undermining GPs’ efforts to promote healthy lifestyles and encourage lifestyle interventions. The accessibility and affordability of healthy foods in the rural environment meant that patients were unable to act on the lifestyle advice given and this made it more challenging for GPs to manage their patients’ lifestyle risks. One GP described this reality in the rural environment:

“High calorie, convenient food is really accessible… it’s drive-thru, so you don’t have to get out of the car, which might suit families or working busy people.” and “Our whole food system and economic system and work system is not well set up for most people.”(GP06)

The need for broader interventions outside healthcare was also identified:

“Until the government makes some serious changes to the way that we market and advertise food, we are never going to fix the metabolic syndrome problem.”(GP11)

##### Socioeconomic Disadvantages in the Rural Setting

GPs noted that patients were frequently at a socioeconomic disadvantage in rural environments and that this disadvantage significantly affected their ability to adhere to risk factor and chronic disease management treatment plans. Financial hardships limited some rural patients’ access to medications, allied health services and lifestyle interventions. One GP observed the following:

“Some of them… haven’t got time [or money] to go to a fancy gym or see a dietitian, so they’re just eating the pills.”(GP01)

GPs highlighted the concerns patients had about their healthcare when it came to costs:

“But because a heart attack is free at the hospitals you can have an expensive gym membership and live a great life that’s going to cost you money. […] Most people will pick the free heart attack or free stroke.”(GP01)

“it’s very difficult to convince people to go to a dietician or an exercise physiologist, let alone spend the extra money to pay someone after those five sessions.”(GP05)

### 3.2. Facilitators

As well as describing the barriers, rural GP participants who were interviewed shared their perspectives on what might assist them in the care and lifestyle management of their patients with metabolic risk for chronic disease in the rural environment.

#### 3.2.1. Systemic Facilitators to Rural Healthcare Delivery

##### Long-Term Relationships with GPs and Rural GP Practices

Continuity of care and trust enabled by sustained doctor–patient relationships emerged as critical facilitators of good chronic disease risk factor management, particularly through lifestyle changes that enhanced patient engagement and adherence.

GP08 emphasised that “the whole thing about chronic diseases is there needs to be regular and sustained attention,” while GP10 described the benefits of this as being the following: “I worked at AMS (Aboriginal Medical Service) here for… almost three years, and in that time, had a nice cohort of patients who I was able to see make that change because of continuity of their primary GP.”

In addition to this, GP02 discussed the value of having a GP over a lifetime: “Because people remember stuff their doctor says and even if you think it’s a throwaway line, you know, I think I was remembering my GP when I was nine years old….You know, it’s little stuff like that can make a difference. So, I think we need to just sow the seed in the patient’s brain, give them positive encouragement when they come in.” (GP02).

Furthermore, the utility of regular reviews in the care plan system, along with the systems in place to facilitate these, was brought to attention as being critical to the ongoing care of patients with risk of chronic disease:

“The powers in the reviews to move the dial. So if you’re not reviewing, it’s like me telling me how to play football, teach you the rules, take you for the first game, then never see you ever again,” and “We’re trying to develop systems where admin, nurse sends, text, email, phone calls to follow up the reviews.”(GP01)

##### Local Multidisciplinary Teams and Interprofessional Communication

Rural GPs found the specific contributions of allied healthcare professionals to improve their patients’ cardiometabolic risk profiles, which is important in terms of their own time constraints and challenges:

“The dietitians are really good at this as they’ve got the time to sit down… look through their shopping basket.”(GP07)

Accessible allied health professionals bolstered care delivery by providing specialised input. Integrated models, which reduced logistical barriers and ensured that patients received holistic support from trusted providers, were praised:

“You need to assemble a team, and you need to collaborate. So it needs co-ordination. And you need to have systems to make it seamless for the patient… that’s exactly what we’ve done at […] and so everything’s under one roof […] You can actually do some studies for every 100 m away from a general practice that the drop-offs occur. So that’s why we’re trying to have a one stop shop.”(GP01)

These teams reduced logistical barriers and ensured that patients received holistic support for lifestyle change from trusted providers.

One GP described their clinic’s services: “The patient is very fortunate with that we have in-house visits from dietician, diabetic educator, renal nurse, podiatrist, Optometry, […], physiotherapists, exercise physiology go to gym.” (GP08)

Effective coordination and communication within multidisciplinary teams were identified by GPs as essential facilitators. GPs reported that seamless communication among providers particularly ensured cohesive care. Good communication and knowing the allied health professional were valued:

“I think for most GPs, you kind of have your favourite allied health people, because you trust them, or you’ve seen good results, or they write you a good letter back. Or you agree with their sort of philosophy and way of practicing.”(GP04)

This GP also described the ideal communication pathway, which they felt rested with the GP at the centre. Robust interprofessional links were needed to bridge the gaps.

“It’s probably going to be the GP who’s actually going to be able to coordinate the care and look after all the other comorbid conditions at the same time.”(GP04)

##### Case Management and Health Coaching

Case management and case managers were promoted as a viable solution to the problem of providing lifestyle care and advice by several GPs in providing structured support and follow-up for those with chronic disease and cardiometabolic risk factors. The benefits of health coaches as case managers were acknowledged, along with the recognition of the costs:

“…[I] actually find health coaches can be really helpful as well, they can be expensive, but they are brilliant. And so I do find if I can, if people can afford a health coach, I will absolutely get them one involved as well. Because I find they can be really helpful with that accountability stuff, which is what makes the lifestyle changes quite challenging.”(GP11)

Another GP highlighted their motivational role:

“I often think people need to see a life coach… that burst of motivation.”(GP05)

These insights suggest case managers can enhance both clinical and logistical aspects of care.

GPs were keen to see money allocated to case management by the government:

“If we identify and allocate more money… into having these case managers… we will be able to identify people who needed assistance with transport and funding and be able to allocate that accordingly.”(GP13)

#### 3.2.2. Broader Facilitators Beyond Healthcare for Chronic Disease and Cardiometabolic Risk in Rural Areas

While GPs’ comments were mostly around the provision of facilitators to chronic disease healthcare delivery, rural GPs were keen to see public health measures that improved the environment for healthy eating for the rural population as well as measures to assist patients financially with access to medical and, specifically, allied health practitioners. GPs advocated for changes at a governmental policy level in terms of addressing socioeconomic and healthcare delivery disparities in the rural community.

“It would imply maybe a society as a whole if, if we make food, good food cheaper and accessible, that would probably have a role. They did this in Scandinavia in the 50s, with a high rate of coronary artery disease. And vegetables and salads… It’s only when the government made plans that every time you bought any food at the supermarket, they had to give you some salads or something like that, which would normally cost quite a bit, but you get it for free. So, in a generation, the whole society of coronary artery disease has just plummeted.”(GP02)

“I think the biggest barrier is Big Food, the food industry. I really do think like, until the government makes some serious changes to the way that we market and advertise food, we are never going to fix the metabolic syndrome problem.”(GP11)

## 4. Discussion

This qualitative study has described rural Western Australian GPs’ perceptions of the barriers and facilitators to management of chronic disease and cardiometabolic risk factors in rural patients, particularly around promoting lifestyle changes. The findings suggest systemic and professional, as well as broader environmental, challenges that hinder the effective management of chronic disease and chronic disease risk factors in the rural environment for Australian GPs.

### 4.1. Systemic Challenges

Systemic challenges dominate the landscape of providing care by GPs for patients with chronic disease and cardiometabolic risk factors, with resource constraints on offering sufficient consultation time, severely limiting GPs’ ability to provide the desired comprehensive lifestyle care for patients with chronic disease metabolic risk factors. The need to keep chronic disease consultations brief is due to the need to attend to other urgent and acute medical conditions in the context of the inadequate numbers of GP providers in rural areas. This restricts the individual GP’s capacity to address the multifaceted needs of patients with chronic disease risk through lifestyle intervention, often resulting in fragmented care focused on acute presentations, rather than preventive strategies.

#### 4.1.1. Financial Disincentives

Financial disincentives under the Australian fee-for-service model of remuneration further discourages GPs providing time-intensive interventions such as lifestyle counselling for risk factor modification [27]. These challenges of lack of time and resources to manage chronic disease have previously been reported for GPs, particularly in relation to nutritional care, but are intensified in a rural setting [28].

#### 4.1.2. GP and Allied Healthcare Shortages

In rural settings, staffing shortages driven by high GP turnover and recruitment difficulties strain primary care resources, with practices being understaffed in terms of GPs and unable to meet patient demand [16]. GPs suggested that the current costs of living were prohibitive in rural areas for many healthcare professionals. Geographic isolation exacerbates these problems, since vast distances and limited transport options for many patients, as is common in rural WA, hinder patients’ access to healthcare facilities and allied health services. Allied health professionals are often able to assist patients with chronic disease and cardiometabolic risk factors best with lifestyle changes, with this being identified as one of the most significant barriers to chronic disease care in a rural setting. GPs reported these challenges being exacerbated by lack of co-location of allied healthcare professionals with each other and with medical services, and with poor referral and provider communication impacting the best care of patients with chronic disease risk factors.

#### 4.1.3. Social and Economic Disadvantages

Further environmental and socioeconomic disadvantages in the rural Australian setting were reported by GPs as compounding the barriers to effective care, restricting patients’ ability to travel to and pay for allied health consultations or afford healthy foods. Despite these many obstacles to chronic disease prevention and risk factor management in the rural environment, particularly with respect to lifestyle management, facilitators suggested by GPs offer potential pathways for the improvement of care. These include the following: (i) Addressing the GP workforce and resource implications—GPs that were interviewed suggested that continuity of care fostered through long-term relationships with patients could help to enable better sustained management of chronic condition risk factors and assist with lifestyle changes for patients. Such continued care relies on there being enough rural GPs and improved models of GP recruitment and retention in rural areas. (ii) GP time constraints—greater time per patient consultation to address chronic disease and risk factor management through lifestyle change requires realistic financial incentives for GPs through improvements in the fee-for-service model of remuneration. (iii) Enhancing multidisciplinary support through multidisciplinary teams, integrating dietitians, exercise physiologists and other allied health professionals, who are ideally co-located geographically to provide comprehensive support while reducing logistical barriers to chronic disease care. Increasing access to relevant allied health professionals was seen as critical to patients’ management of chronic disease risk factors. (iv) Case management and lifestyle coaching through case managers and (v) environmental and public health interventions—GPs wanted reform in public health policy in rural areas around promoting healthy environments for rural patients, especially about healthy foods and combatting socioeconomic disadvantages.

### 4.2. Implications for Practice and Policy

The findings suggest several practical strategies to enhance chronic disease care and the management of chronic disease metabolic risk factors through addressing lifestyle change in rural Western Australia and Australian rural communities more broadly. Policy interventions for key issues are essential to address systemic barriers and leverage facilitators, aligning with the study’s call for reducing rural health inequities:

#### 4.2.1. GP Workforce Shortages

The GPs noted an ongoing workforce shortage of GPs, with poor levels of retention in rural WA. A consequence of this is difficulty providing comprehensive care to patients with chronic disease and cardiometabolic risk factors, particularly through lifestyle change, which requires a significant time investment. GPs recognised that continuity of care and developing long-term relationships with patients is a key component of chronic disease care to provide the best care through lifestyle intervention. While fly-in–fly-out and drive-in–drive-out services [29] and telehealth are used to support care [30], these often do not serve to develop local healthcare infrastructure and do not contribute to establishing a therapeutic relationship or continuity of care. Incentivising resident doctors in rural towns is the least expensive healthcare model [31] and this can be facilitated through evidence-based retention strategies such as infrastructure provision, competitive renumeration, sustainable workplace organisation, a healthy professional environment and community support [32].

Most of the GPs in our study did not believe the current fee-for-service model was adequate for chronic disease and cardiometabolic risk care needs, generally or in the rural environment. Adequate renumeration for longer consultations is advocated for by the Royal Australian College of General Practitioners [33]. The Medicare Benefits Scheme has undergone significant reform to its chronic disease care model on the 1st of July 2025, with the outcomes to be reviewed [34].

#### 4.2.2. Access to Allied Healthcare

Poor access to allied healthcare professionals to assist with addressing lifestyle concerns in patients with chronic disease and cardiometabolic risk factors was another problem identified. Having the necessary allied health and nursing staff co-located geographically has been advocated for as being ideal to provide adequate chronic disease care through access and improved interprofessional communication.

#### 4.2.3. Coordination of Care

GPs reported the need for better coordinated care for those with chronic disease and cardiometabolic risk factors, proposing case managers who can help manage and advocate for the patient. For example, registered nurses leading case management can assist through improved communication and accessibility to health resources [35], being a patient advocate, educating the patient and providing emotional support [36], and they have been shown to improve psycho-behavioural outcomes and the quality of life of patients where they have acted as case managers [37]. Case management needs further exploration as a model for chronic disease care in Australia, including assessment of the clinical and health economic costs of bringing this model into rural healthcare.

#### 4.2.4. Lifestyle Coaching

Lifestyle coaching was another intervention suggested by the rural GPs so that patients could be assisted with managing their chronic disease and cardiometabolic health through coaching beyond the usual medical care they received. A number of systematic reviews [38,39] have suggested that health coaching by nurses or GPs, either outside of medical care or integrated into primary care, can lead to improved outcomes for patients [38]. These interventions have shown promise in developing self-management in patients—for example, in T2DM—although further investigation is required [38]. In Geraldton, the Integrated Chronic Disease Care Program offers health coaching for individuals with diabetes, chronic heart disease or chronic respiratory disease but availability is restricted and limited to those with socioeconomic disadvantages [40]. The PHYZ X 2U model provides health coaching and access to allied healthcare services in rural populations through telehealth and shows feasibility in helping participants achieve chronic disease goals [41]. Life coaching models in relation to rural primary care for chronic disease and cardiometabolic risk factors require further exploration.

#### 4.2.5. Unhealthy Environments

Some of the GPs interviewed in this study reported unhealthy environments for chronic disease prevention and cardiometabolic risk management in rural WA. The GPs unanimously advocated for reform in public health policy in rural areas. Low access to healthy environments increases non-communicable diseases and can impact the health system’s resource cost significantly. Indeed, healthy food choices in rural Australia are limited, with fresh foods being expensive due to a paucity of large retail store chains in rural towns, and with rural pubs and cafes frequently only providing easy access to high-fat foods, undermining dietary recommendations and fostering an obesogenic environment that challenges both patients and GPs [42,43]. Swinburn et al.’s INFORMAS framework, which globally monitors food environments to reduce obesity and non-communicable diseases and improve healthy food environments, advocates for such measures to shift dietary patterns through governmental intervention [44]. Advocacy for governmental policy reform to incentivise the provisioning of healthy foods should be considered, such as subsidies on healthy foods or taxes on sugar-sweetened beverages.

### 4.3. Strengths and Limitations

This study’s strengths include its in-depth qualitative nature, providing nuanced insights into rural GPs’ experiences, which enrich the understanding of chronic disease risk prevention in the context of cardiometabolic risk factors in underserved regions. The critical qualitative approach highlights systemic inequities that are often overlooked in clinical research and gives a voice to rural GPs. The lead investigator’s role as a rural GP enhanced the study’s contextual relevance, informing the design and interpretation of the findings.

Limitations include the small sample size (15 GPs), which, while typical for qualitative research which seeks in-depth understanding rather than representativeness, does require caution in its generalisability to other rural contexts or broader GP populations. The focus on GP perspectives excludes those of patients, allied health professionals and policymakers, potentially limiting the scope of insights in the care ecosystem. Convenience and snowball sampling may introduce selection bias, favouring GPs with strong opinions on chronic disease care. The regional specificity to Western Australia may not fully translate to other rural settings with differing healthcare systems or cultural dynamics.

## 5. Conclusions

Barriers to managing cardiometabolic chronic disease risk through lifestyle interventions in rural Western Australia include healthcare resource constraints, geographic isolation, socioeconomic disadvantages and a disease-promoting food environment. However, there are facilitators—continuity of care, local multidisciplinary teams, case management and interprofessional communication—which offer practical solutions to enhance care delivery. While some of these barriers and solutions have previously been identified in urban environments and in other Australian and international rural settings, rural GPs in WA considered additional solutions such as lifestyle coaching and case management to be particularly important and necessary in the WA setting. This likely reflects aspects of WA’s remoteness, the limited availability of allied health staff and the need for GPs to adopt less traditional roles in caring for their patients with chronic disease and cardiometabolic risk, relinquishing some support to other providers while taking a broader coordinating role. To achieve this requires a capable and available allied health workforce.

Addressing the identified challenges requires a multifaceted approach, integrating clinical practice improvements, policy reforms and community-based interventions. By prioritising system changes through revised funding models, workforce development and leveraging facilitators like telehealth and multidisciplinary care, we can reduce rural health inequities and improve outcomes for patients with complex chronic diseases. Future research should validate these findings through quantitative studies, intervention trials and broader stakeholder and patient perspectives to inform evidence-based strategies for rural chronic disease care.

## Figures and Tables

**Figure 1 healthcare-14-00113-f001:**
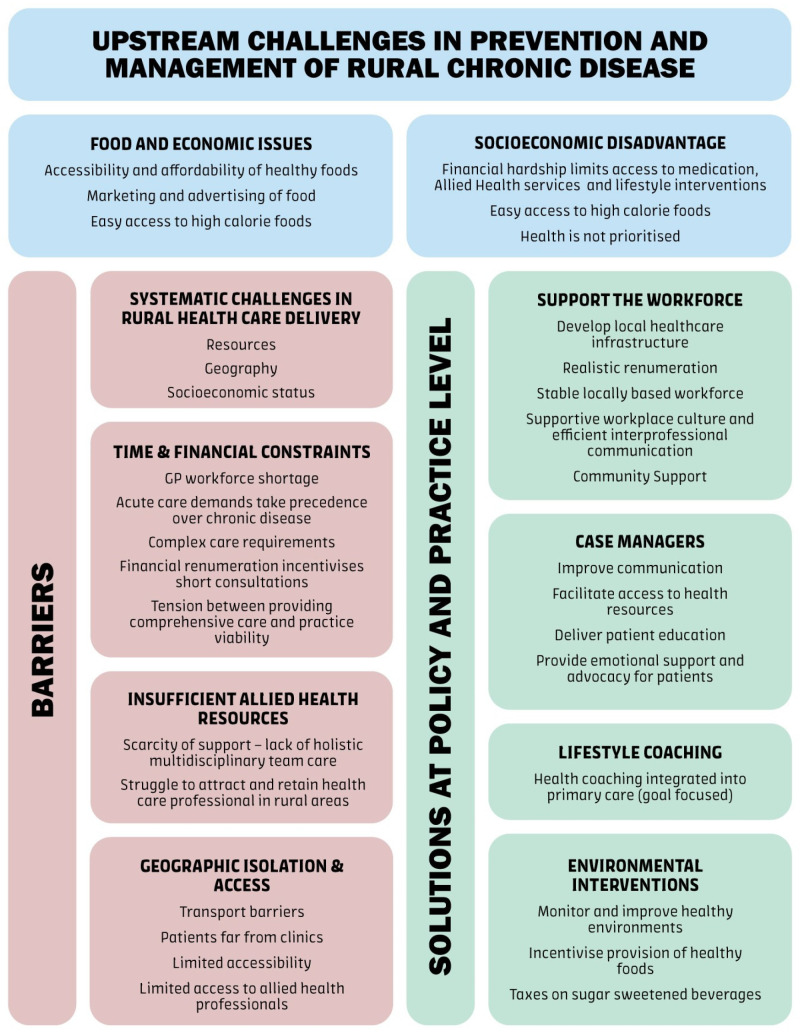
Barriers and facilitators to chronic disease care in rural general practice.

## Data Availability

To protect the confidentiality of the research participants, the data generated by this study is not publicly available. However, interested parties can contact the corresponding author with respect to further access to the data.

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
