# Peer review of "Rural General Practitioners’ Perceptions of the Barriers and Facilitators of Chronic Disease and Cardiometabolic Risk Factor Care Through Lifestyle Management—A Western Australian Qualitative Study"

_healthcare, 2026, doi:10.3390/healthcare14010113_

Round 1
Reviewer 1 Report
Comments and Suggestions for Authors
Thank you for this important and timely submission. The qualitative approach which provides nuanced insights into the lived experiences of rural GPs allows for the exploration of complex systemic and environmental challenges that quantitative studies often overlook. The qualitative thematic analysis is appropriate and well executed. Please see the following suggestions to strengthen the manuscript:
- Title - Consider revising the title to better indicate the study design. Perhaps adding the phrase ‘A Qualitative Study’ will improve clarity for readers.
- Abstract - Consider revising the methods section by removing details (e.g., recruitment networks, software used) and replacing this with more information on the design, sample size, and analysis approach. Consider making the conclusion more impactful by highlighting practical implications for policy and practice.
- Introduction – Toward the end of the first paragraph consider better highlighting the research gap (i.e., contrasting rural versus urban studies and clarifying why Western Australia is a unique context).
- Methods - The qualitative design and thematic analysis is appropriate for exploring perceptions of barriers and facilitators. The use of Braun and Clarke’s framework is well justified. The Methods section could be improved by adding more detail on the description of transcription and coding processes. Consider mentioning the essential steps rather than listing software tools and consider adding a brief statement on how trustworthiness was ensured (e.g., triangulation, member checking, or peer debriefing). Consider adding subheadings for recruitment, data collection, and analysis to enhance readability.
- Results – The results are presented clearly (barriers and facilitators) and supported by participant quotes. Figure 1 is helpful and comprehensive.
- Discussion and Conclusions – The discussion is comprehensive but somewhat lengthy. Consider using subheadings to enhance readability. Also, consider adding a statement to emphasize what is new about this study. For example, the identification of facilitators such as case management and health coaching. These insights add value to rural health literature. Consider noting that future research may focus on the value of including the patient's perspective.
Author Response
Thank you for your comments regarding our manuscript. See attached file for updated manuscript.
- Title - Consider revising the title to better indicate the study design. Perhaps adding the phrase ‘A Qualitative Study’ will improve clarity for readers.
The title has now been changed to: Rural General Practitioners’ Perceptions to the Barriers and Facilitators of Chronic Disease and Cardiometabolic Risk Factor Care through Lifestyle Management – A Western Australian Qualitative Study
- Abstract - Consider revising the methods section by removing details (e.g., recruitment networks, software used) and replacing this with more information on the design, sample size, and analysis approach. Consider making the conclusion more impactful by highlighting practical implications for policy and practice.
The abstract has now been revised according to the feedback of the reviewer.
- Introduction – Toward the end of the first paragraph consider better highlighting the research gap (i.e., contrasting rural versus urban studies and clarifying why Western Australia is a unique context).
The research gap has been highlighted and the unique Western Australian context has now been further described in the introduction.
- Methods - The qualitative design and thematic analysis is appropriate for exploring perceptions of barriers and facilitators. The use of Braun and Clarke’s framework is well justified. The Methods section could be improved by adding more detail on the description of transcription and coding processes. Consider mentioning the essential steps rather than listing software tools and consider adding a brief statement on how trustworthiness was ensured (e.g., triangulation, member checking, or peer debriefing). Consider adding subheadings for recruitment, data collection, and analysis to enhance readability.
Subheadings have now been added to the methods section.
A short paragraph has been added regarding the coding process.
Trustworthiness is now addressed.
- Results – The results are presented clearly (barriers and facilitators) and supported by participant quotes. Figure 1 is helpful and comprehensive.
Thank you for your comments regarding the results and figure
- Discussion and Conclusions – The discussion is comprehensive but somewhat lengthy. Consider using subheadings to enhance readability. Also, consider adding a statement to emphasize what is new about this study. For example, the identification of facilitators such as case management and health coaching. These insights add value to rural health literature. Consider noting that future research may focus on the value of including the patient's perspective.
Thank you for your comments.
Headings have now been added to the discussion section
A statement is added about what is new in the study with respect to rural versus urban needs in the conclusion of the manuscript.
The final sentence has been clarified to include patients as well as stakeholders’ perspectives in future research.

Reviewer 2 Report
Comments and Suggestions for Authors
Authors summarize barriers to chronic disease and cardiometabolic risk-factor management in rural Western Australia (WA). The manuscript has strong real-world relevance, with clear potential to inform rural health policy and service planning. The results are detailed, and the discussion offers thoughtful policy implications.
Suggestions for consideration
-
Strengthen the introduction with a concise literature framing.
Consider adding a brief synthesis of the current literature on healthcare access in rural Australia, particularly for chronic disease and cardiometabolic prevention/treatment (e.g., primary care access, workforce shortages, travel burden, continuity of care, affordability, culturally safe care for Aboriginal communities, and access to allied health and diagnostics). Then explicitly state why WA may differ from prior literature (e.g., geography/remoteness, service distribution, mining/industry-driven population mobility, local workforce models, or distinct referral pathways). This would better motivate the study objective and help readers contextualize the setting. -
Clarify contribution by comparing findings to prior evidence.
In the Discussion, please summarize how your key findings align with or diverge from existing evidence from other Australian regions (and, where relevant, comparable international rural settings). A short “what is consistent vs what is new/WA-specific” comparison would make the manuscript’s incremental contribution more explicit and improve transferability of insights for readers outside WA. -
Tighten and refocus policy implications.
The policy implications section is valuable but currently lengthy. Consider streamlining to a small number of high-level, actionable recommendations (e.g., 3–5) linked directly to the main findings, and move granular details or descriptive content back into the Results (or an appendix/supplement, if appropriate). This will improve readability and strengthen the narrative flow.
Author Response
Authors summarize barriers to chronic disease and cardiometabolic risk-factor management in rural Western Australia (WA). The manuscript has strong real-world relevance, with clear potential to inform rural health policy and service planning. The results are detailed, and the discussion offers thoughtful policy implications.
Suggestions for consideration
- Strengthen the introduction with a concise literature framing.
Consider adding a brief synthesis of the current literature on healthcare access in rural Australia, particularly for chronic disease and cardiometabolic prevention/treatment (e.g., primary care access, workforce shortages, travel burden, continuity of care, affordability, culturally safe care for Aboriginal communities, and access to allied health and diagnostics). Then explicitly state why WA may differ from prior literature (e.g., geography/remoteness, service distribution, mining/industry-driven population mobility, local workforce models, or distinct referral pathways). This would better motivate the study objective and help readers contextualize the setting.
Thank you to the reviewer for their comments. The above points have been further addressed in the introduction.
- Clarify contribution by comparing findings to prior evidence.
In the Discussion, please summarize how your key findings align with or diverge from existing evidence from other Australian regions (and, where relevant, comparable international rural settings). A short “what is consistent vs what is new/WA-specific” comparison would make the manuscript’s incremental contribution more explicit and improve transferability of insights for readers outside WA.
Thank you this has been highlighted in the conclusion of the paper.
- Tighten and refocus policy implications.
The policy implications section is valuable but currently lengthy. Consider streamlining to a small number of high-level, actionable recommendations (e.g., 3–5) linked directly to the main findings, and move granular details or descriptive content back into the Results (or an appendix/supplement, if appropriate). This will improve readability and strengthen the narrative flow.
Thank you for this suggestion. We have edited this section to tighten it but have not taken up the reviewer’s suggestion to limit the number of recommendations. Comprehensive changes are needed if chronic disease care is to be improved for rural patients. Also, our discussion is summarised in the figure which highlights the need for multilevel reforms and this correlates with our findings as reported in the results. As per Reviewer 1’s comment, we have added subtitles to improve readability.